# Facilitated Photocatalytic Degradation of Rhodamine B over One-Step Synthesized Honeycomb-Like BiFeO_3_/g-C_3_N_4_ Catalyst

**DOI:** 10.3390/nano12223970

**Published:** 2022-11-10

**Authors:** Haoran Cui, Zhipeng Wang, Guoqiang Cao, Yiwan Wu, Jian Song, Yu Li, Le Zhang, Jiliang Mu, Xiujian Chou

**Affiliations:** 1School of Chemical Engineering and Technology, North University of China, Taiyuan 030051, China; 2School of Instrument and Electronics, North University of China, Taiyuan 030051, China; 3State Key Laboratory of Coal Conversion, Institute of Coal Chemistry, Chinese Academy of Sciences, Taiyuan 030001, China; 4School of Food Science and Engineering, Wuhan Polytechnic University, Wuhan 430023, China

**Keywords:** one-step methodology, BiFeO_3_/g-C_3_N_4_, honeycomb-like morphology, Z-scheme heterojunction, ferroelectric

## Abstract

In the present work, a facile one-step methodology was used to synthesize honeycomb-like BiFeO_3_/g-C_3_N_4_ composites, where the well-dispersed BiFeO_3_ strongly interacted with the hg-C_3_N_4_. The 10BiFeO_3_/hg-C_3_N_4_ could completely degrade RhB under visible light illumination within 60 min. The degradation rate constant was remarkably improved and approximately three times and seven times that of pristine hg-C_3_N_4_ and BiFeO_3_, respectively. This is ascribed to the following factors: (1) the unique honeycomb-like morphology facilitates the diffusion of the reactants and effectively improves the utilization of light energy by multiple reflections of light; (2) the charged dye molecules can be tightly bound to the spontaneous polarized BiFeO_3_ surface to form the Stern layer; (3) the Z-scheme heterojunction and the ferroelectric synergistically promoted the efficient separation and migration of the photogenerated charges. This method can synchronously tune the micro-nano structure, surface property, and internal field construction for g-C_3_N_4_-based photocatalysts, exhibiting outstanding potential in environmental purification.

## 1. Introduction

With the rapid development of contemporary civilization, people’s material standard of life has been significantly improved. Meanwhile, ever-worsening environmental problems, such as drinking water safety, have become a common concern for all humans. Dyes are typical pollutants that are released into natural water sources, and they contain a large number of toxins, mutagens, and carcinogens [1,2]. In particular, Rhodamine B (RhB) is an extensively used cationic xanthene dye and possesses long-term hazards for human health [3,4]. Though it is also a well-liked biological stain in biomedical research attributed to its adaptability, degradation of RhB remains crucial and necessary [5]. Conventional RhB degradation using the adsorption methods results in secondary pollution, whereas RhB degradation using biochemical processes is still challenging because of its chemically persistent [6,7]. Advanced techniques and materials regarding the degradation of RhB are still under investigation and have attracted more attention.

Heterogeneous photocatalytic technology is considered to be one promising method to resolve RhB pollution concerns, attributed to its low-cost, safe, efficient, and eco-friendly characteristics [8,9]. Among many photocatalytic materials, g-C_3_N_4_ has received widespread attention from academia and industry. As a new non-metallic visible light-driven catalyst, it possesses the advantages of being cheap and easy to obtain and non-toxic and having a narrow band gap and a high conduction band position. Nonetheless, technical short slabs, such as the high recombination rate of photogenerated carriers, low surface area, and limited redox potential, restrict the application of g-C_3_N_4_ [10,11,12]. Multiple strategies have been extensively developed to improve the photocatalytic performance of g-C_3_N_4_, including micro-nano structure control, defect regulation impurity doping, and heterostructure engineering [13]. The essence of these strategies is to improve the separation of excitation charge, inhibit the recombination of carriers, and adjust the wavelength range of absorbable light [14,15,16,17,18]. 

The internal electric field, which is usually formed inside or at the interface of semiconductors, profoundly influences carrier migration behavior. One of the best-known examples is the fabrication of heterostructures between g-C_3_N_4_ and other semiconductors, which has been revealed to be an effective route to inhibit the recombination of photogenerated carriers [19]. Various g-C_3_N_4_-based composite photocatalysts have been prepared for the degradation of organic pollutants, such as TiO_2_-g-C_3_N_4_, ZnO-g-C_3_N_4_, BiFeO_3_-g-C_3_N_4_, Fe_2_O_3_-g-C_3_N_4_, Bi_2_WO_6_-g-C_3_N_4_, BiVO_4_-g-C_3_N_4_, and TaON-g-C_3_N_4_. These photocatalysts have enhanced catalytic performance compared to single-component materials [20,21,22,23]. Among them, as a typical perovskite structure (ABO_3_) material, BiFeO_3_ can be activated by visible light. It has a relatively narrow band gap (2.1–2.7 eV) due to the valence band consisting of Bi 6s and O 2p hybrid orbitals [24]. The magnetization derived from the Fe^3+^ ion facilitates the separation of catalysts from the liquid phase. Moreover, BiFeO_3_ is a typical ferroelectric material [25,26]. The displacements of individual atoms or ions in the crystal structure result in internal polarization. The internal polar regions favored driving photogenerated carriers to move in opposite directions; therefore, it inhibits the recombination and enhances photocatalytic efficiency [27]. BiFeO_3_ has been successfully combined with g-C_3_N_4_ to form heterojunctions and applied to the degradation of soluble organic pollutants [23,28]. The g-C_3_N_4_/BiFeO_3_ composites exhibit higher visible light catalysis activity than the pure g-C_3_N_4_ or BiFeO_3_, mainly due to the improved charge separation efficiency. The general preparation method of g-C_3_N_4_/BiFeO_3_ is a relatively complex process; it needs to synthesize the g-C_3_N_4_ and BiFeO_3_ powders individually and then recombine them to form heterojunctions. In addition, most reported catalysts are irregular bulk/sheet-like materials with limited light energy utilization. Therefore, developing an easy and effective preparation method is necessary to construct the interfaces between g-C_3_N_4_ and BiFeO_3_ and tune the texture and structure precisely to further improve the photocatalytic performance.

In this work, BiFeO_3_ particles decorated with honeycomb-like graphite C_3_N_4_ are constructed by a facile one-step method. The degradation of Rhodamine B was used to evaluate the photocatalytic performance of the catalysts. The BiFeO_3_/gh-C_3_N_4_ morphology, adsorption properties, and the formation of the internal field were discussed in detail.

## 2. Experimental

### 2.1. Synthesis of the Photocatalysts

The composite catalysts were prepared using a one-step method. Typically, equivalent molars of Bi(NO_3_)_3_·5H_2_O and Fe(NO_3_)_3_·9H_2_O were dissolved in 20 mL of deionized water along with 15 g of urea. The solution was placed in a crucible and kept in a muffle furnace, followed by calcination at 673 K for 1 h and then at 723 K for another 1 h. The calcined samples were named *x*BiFeO/hg-C_3_N_4_ (*x* = 5, 10, 20), where *x* represents the BiFeO_3_ loading. The synthesis of honeycomb-like g-C_3_N_4_ (labeled hg-C_3_N_4_) was similar to the *x*BiFeO/hg-C_3_N_4_ samples, whereas Bi(NO_3_)_3_·5H_2_O and Fe(NO_3_)_3_·9H_2_O were not added. The yield of hg-C_3_N_4_ is about 0.25 g after 15 g urea calcination. For comparison, the bulk g-C_3_N_4_ (labeled bg-C_3_N_4_), BiFeO_3_ (labeled BiFeO), and 10BiFeO/bg-C_3_N_4_ were also prepared following a similar procedure [23]. All of the chemicals, including urea, Fe(NO_3_)_3_·9H_2_O, and Bi(NO_3_)_3_·5H_2_O were commercially purchased, analytical grade, and used without any further purification.

### 2.2. Catalysts Characterizations

The X-ray diffraction (XRD) patterns were carried out on a Rigaku D/Max-2000 instrument using monochromatic Cu Ka radiation (k = 1.5418 Å) to analyze the crystallinity. X-ray photoelectron spectroscopy (XPS) measurements were taken on a photoelectron spectrometer using Al Ka radiation as the excitation source (Thermo Fisher ESCALAB 250Xi, Waltham, MA, USA). High-resolution transmission electron microscopy (HRTEM) was performed at 200 kV with a JEM-2100F (Osaka, Japan) to collect the structural information of the products. The time-resolved PL spectra were obtained on the Steady State and Transient State Fluorescence Spectrometer (Edinburgh FLS980, Livingston, UK). The excitation wavelength of PL analysis was 375 nm, and the emission wavelength of the TRPL analysis was 540 nm. The M-H loop measurements were carried out at room temperature on a SQUID VSM DC magnetometer (San Diego, CA, USA). The electron spin resonance was performed on an electron paramagnetic resonance spectrometer (ESR, Bruker-A300, Middlesex, MA, USA). The measurement was prepared by adding the samples in a 50 mM DMPO solution with aqueous dispersion for DMPO-•OH and methanol dispersion for DMPO-•O_2_^−^. UV-vis diffuse reflectance spectra (DRS) were obtained by a PE Lambda 950 spectrometer with BaSO_4_ as a reference. N_2_-physisorption analysis was measured on a Micromeritics ASAP-2020 apparatus (Norcross, GA, USA). Before the adsorption analysis, the fresh and used samples were degassed at 150 °C for 5 h and 60 °C for 24 h, respectively. The specific surface area was obtained from the 5-point Brunauer–Emmett–Teller (BET) procedure. The average pore diameter and pore volume were determined by the Barrett–Joyner–Halenda (BJH) method. Fourier-transform infrared (FT-IR) spectra were recorded on a Bruker Tensor 27 Fourier-transform infrared spectrometer at room temperature by the subtraction of the background reference. One milligram of each powder sample was diluted with 100 mg of vacuum-dried IR-grade KBr. Then, 16 scans were collected for each sample at a resolution of 4 cm^−1^. The photoelectrochemical (PEC) performances of the prepared photoanodes were recorded on an electrochemical workstation (CHI 660D Instruments, Austin, TX, USA) with a standard three-electrode system (a saturated calomel electrode (SCE) reference electrode and a Pt foil counter electrode). Then, 0.5 M NaSO_4_ solution was used as the electrolyte. The measured potential vs. SCE was converted to a normal hydrogen electrode (NHE) scale using the Nernst equation: E_NHE_ = E_SCE_ + 0.2412. A 300 W Xe lamp (Beijing, China) was used as a light source. Transient photocurrent measurements at a constant bias (0.5 V) with chopped illumination were also conducted to examine the steady-state photocurrent densities of the photoanodes. The electrochemical impedance spectra (EIS) were carried out in the frequency range of 0.05–10^5^ Hz. The electrochemical impedance spectrum (EIS) was measured using a CHI 660D electrochemical workstation (Austin, TX, USA). The working electrode was prepared according to the following process: 20 mg of the as-prepared sample was suspended in 0.5 mL of DMF, which was then dip-coated on a 10 mm × 20 mm indium–tin-oxide (ITO) glass electrode. The electrode was then annealed at 350 °C for 1 h at a heating rate of 6 °C min^−1^. EIS was carried out at an open-circuit potential. A sinusoidal ac perturbation of 5 mV was applied to the electrode over the frequency range of 0.05–10^5^ Hz.

### 2.3. Photocatalytic Activity Test

The photocatalytic degradation of Rhodamine B (RhB, Sigma, 99.99%, Balcatta, WA, USA) was performed under a simulated solar light source using a 300 W xenon lamp (Beijing, China). Then, 50 mg of the photocatalysts were mixed into 100 mL of RhB solution (10 mg/L). Prior to the photocatalytic reaction, the mixed solution was stirred magnetically in a dark room for 30 min to make the catalyst uniformly dispersed. It also ensured that the dye molecules reached an adsorption/desorption equilibrium on the catalyst surface. The mixture was subjected to a photocatalytic reaction under simulated sunlight. The concentration of the mixed solution was tested in intervals at the same time, and a small amount of the reaction solution was centrifuged to remove the catalyst particles to obtain the supernatant. The remaining concentration of the RhB dye was measured by UV-Vis spectrophotometer with a test wavelength of 554 nm. The photocatalyst in the reaction solution was recycled by centrifuging at 6000 rpm/min for 10 min, washing with distilled water, and drying at 75 °C for 10 h. The collected photocatalyst was used to degrade the RhB solution again and performed the same cycle three times to evaluate stability and recyclability.

## 3. Results and Discussion

### 3.1. Structure and Morphology

Figure 1 displays the XRD patterns of pure hg-C_3_N_4_ and *x*BiFeO/hg-C_3_N_4_ catalysts, along with bg-C_3_N_4_ and 10BiFeO/bg-C_3_N_4_ catalysts for comparison. For the pure hg-C_3_N_4_ and bg-C_3_N_4_ samples in Figure 1a, the diffraction peaks at 12.84° and 27.36° are assigned to the (100) and (002) planes of graphite C_3_N_4_ (JCPDS 87-1526), due to in-plane structural packing motif and interlayer stacking [29,30], respectively. The characteristic diffraction peaks of hg-C_3_N_4_ shifted to lower diffraction angles and became weaker, which was caused by the introduction of H_2_O, resulting in a lower degree of condensation of the precursors during the heat treatment [25]. For the 10BiFeO/hg-C_3_N_4_ and 10BiFeO/bg-C_3_N_4_ samples, the peaks that were observed at 22.41°, 31.75°, 32.07°, 38.95°, 39.48°, 45.75°, 51.31°, and 51.74° are assigned to the (012), (104), (110), (006), (202), (024), (116), and (122) planes of BiFeO_3_ (JCPDS 86-1518). In addition, the characteristic diffraction peak at 27.36° of graphite C_3_N_4_ is also observed, indicating that BiFeO_3_ and graphite C_3_N_4_ coexist in all the composite catalysts. The content of BiFeO_3_ was determined by ICP analysis and was almost equal to the theoretical value (Table 1). Compared with 10BiFeO/bg-C_3_N_4_, the intensity of the BiFeO_3_ characteristic diffraction peaks significantly decreases, suggesting the good dispersion of BiFeO_3_ particles on the hg-C_3_N_4_. Moreover, the intensity of the BiFeO_3_ characteristic diffraction peaks increased sharply as the content of BiFeO_3_ rose from 5% to 20% (Figure 1b). This further confirmed the abundance of BiFeO_3_ while its loading increased.

The FT-IR transmission spectra were employed to gain further insight into the structural features of the samples, as shown in Figure 2. For the pure hg-C_3_N_4_ and bg-C_3_N_4_ samples (Figure 2a), the multiple peaks located at 1650–1200 cm^−1^ were attributed to the stretching of the C-N bond and C-N heterocycles in aromatics. The peak at around 808 cm^−1^ represented the out-of-plane bending vibration of tri-s-triazine units [31,32]. The broad peak in 3000–3500 cm^−1^ was attributed to the stretching of the NH_x_ groups [31]. The stronger peak at 3000–3500 cm^−1^ of hg-C_3_N_4_ suggests a lower degree of condensation, which is consistent with the XRD results. For the composite samples, as shown in Figure 2b, M-O (Fe-O or B-O) vibration peaks appeared below 600 cm^−1^ in addition to the peaks of graphite C_3_N_4_ [33]. The intensity of the NH_x_ groups on 10BiFeO/hg-C_3_N_4_ was much weaker than that of the 10BiFeO/bg-C_3_N_4_ sample. As aforementioned, the BiFeO_3_ particles have good dispersion on hg-C_3_N_4_. This could be mainly ascribed to the NH_x_ groups on the hg-C_3_N_4_, which anchor the metal species and facilitate the dispersion [34,35].

The synthesized samples were characterized by TEM to demonstrate the morphological features. In Figure 3a, large particle stacking can be observed in the bg-C_3_N_4_ sample. In comparison, the hg-C_3_N_4_ sample is assembled with a cellular lamellar structure with abundant pore structures distributed on the lamellar surface. In the synthesis process, the thermal decomposition of urea generated bubbles on the sample surface. The introduction of H_2_O caused the bubbles to break on the surface and, eventually, formed pore structures [36]. It can be clearly seen from Figure 3d that the BiFeO_3_ particles were more uniformly dispersed and less aggregated on hg-C_3_N_4_ compared with those on bg-C_3_N_4_ (Figure 3c), which are in good agreement with the XRD results.

The N_2_-physisorption analysis was measured to investigate the texture property of the synthesized catalysts. According to the IUPAC recommendation, all of the samples in Appendix A showed typical type IV isotherms and H_3_-shaped hysteresis loops, which mainly originated from slit-like pores formed by the accumulation of bg-C_3_N_4_ particles or hg-C_3_N_4_ cellular layers [37]. The texture parameters of all of the samples are listed in Table 1. The surface area of the hg-C_3_N_4_ and 10BiFeO/hg-C_3_N_4_ were higher than bg-C_3_N_4_ and 10BiFeO/bg-C_3_N_4_. With the increase in BiFeO loading, the surface area of the *x*BiFeO/hg-C_3_N_4_ samples gradually decreased.

XPS was carried out to determine the surface chemical compositions and the interaction between BiFeO_3_ and graphite C_3_N_4_. Figure 4a displays the full XPS spectra of the obtained composite samples. The survey results confirmed the existence of Bi, Fe, O, N, and C elements [38]. In Figure 4b, the XPS spectra of Bi 4f, and the peaks located at 164.0 eV and 158.7 eV, were ascribed to characteristic Bi 4f_5/2_ and Bi 4f_7/2_, which implied the existence of Bi^3+^ species [39,40]. The Bi 4f peaks of 10BiFeO/hg-C_3_N_4_ shifted to lower binding energy, indicating the strong interaction between BiFeO_3_ and hg-C_3_N_4_. The peaks centered at 710.6 eV and 724.2 eV in the Fe 2p spectrum are assigned to Fe^3+^ (Figure 4c). The O 1s spectrum of the composite samples showed three peaks located at 529.6 eV, 532.2 eV, and 531.1 eV, corresponding to Bi/Fe-O (BiFeO_3_), C-O-C, and C=O (Figure 5d), respectively. The peaks at 284.8 eV and 288.2 eV in C 1s spectra are attributed to C-C coordination and the N=C-N_2_ (Figure 4e). As shown in Figure 4f, the N 1s spectrum can be fitted into three peaks centered at 398.7 eV, 399.5 eV, and 401.1 eV, originating from sp^2^-hybridized nitrogen (C-N-C), sp^3^-hybridized nitrogen (N-(C)_3_), and amino functional C-NH_x_, respectively. These three N 1s peaks of 10BiFeO/hg-C_3_N_4_ shifted to higher binding energy, and the intensity of the C-NH_x_ peak became very weak [41]. This is in line with the FTIR results that fewer NHx were identified on the 10BiFeO/hg-C_3_N_4_ surface. Based on the above analysis, the composites consisting of BiFeO_3_ and graphite C_3_N_4_ were successfully synthesized, and the strong interaction between BiFeO_3_ and hg-C_3_N_4_ existed in 10BiFeO/hg-C_3_N_4_.

### 3.2. Evaluation of Photocatalytic Performance

The photocatalytic performance of the as-synthesized samples was evaluated by the photodegradation of RhB dye under xenon lamp irradiation. As shown in Figure 5a, RhB self-degradation without any photocatalysts can be ignored. For the pure graphite C_3_N_4_ samples, the honeycomb-like g-C_3_N_4_ showed a higher photocatalytic degradation capacity (74%) than the bulk g-C_3_N_4_ (58%) after 60 min of illumination. The better performance of hg-C_3_N_4_ mainly resulted from the higher photoenergy efficiency, reduced surface defects, and enhanced charge separation [36]. While the pure BiFeO_3_ only exhibited 43% photodegradation of RhB. The photocatalytic activity of the composite samples was better than pure graphite C_3_N_4_ and BiFeO_3_. The formation of heterojunction and the micro-nanostructure of the BiFeO/hg-C_3_N_4_ samples significantly impact photocatalytic activity. The 10BiFeO/hg-C_3_N_4_ possesses the best photocatalytic degradation capacity. Nearly 100% of RhB is degraded within 60 min. We used the Langmuir–Hinshelwood pseudo-first-order kinetics model to calculate the degradation rate constant k [42]. As shown in Figure 5b, the constant k of 10BiFeO/hg-C_3_N_4_ was approximately three times that of pristine hg-C_3_N_4_ and seven times that of BiFeO_3_. Meanwhile, the physical mixture of BiFeO_3_ and pure graphite C_3_N_4_ did not show excellent photocatalytic activity. This further suggested that the formation of heterojunction plays a critical role in improving photocatalytic performance.

For the *x*BiFeO/hg-C_3_N_4_ samples (Figure 5c), with the increasing amounts of BiFeO_3_ loading, the degradation activity displayed the trend of volcano type, and the optimal loading was 10%. A limited number of heterojunctions were formed when the BiFeO_3_ loading was low. BiFeO_3_ tends to agglomerate and hinder the effective formation of heterojunctions when excessive BiFeO_3_ is loaded. As a typical ferroelectric material, the built-in electric field of BiFeO_3_ could be influenced by external electrical stimuli. Therefore, we also investigate the effects of electricity on photocatalytic performance. In Figure 5d, compared with the virgin counterpart, electrically poled 10BiFeO/hg-C_3_N_4_ exhibited a better degradation rate. This could be attributed to the enhanced build-in electric field caused by the remanent polarization after the poling process. The poling process acted as an internal driving force to separate and migrate the photogenerated carriers [43].

The cycling runs for the degradation of RhB were performed under the same conditions to evaluate the stability and recyclability of the 10BiFeO/hg-C_3_N_4_. Figure 6a illustrates the degradation efficiencies of RhB catalyzed by 10BiFeO/hg-C_3_N_4_ in three straight cycles. No apparent decline in the photocatalytic activity was observed during the three cycles, indicating the good stability of the 10BiFeO/hg-C_3_N_4_. Figure 6b shows the magnetic hysteresis loops of the used catalysts. The inset picture displays that the used catalysts can be easily separated by the external magnetic field, which could promote the application of the composites.

### 3.3. Reaction Mechanisms

As aforementioned, the photodegradation of RhB over 10BiFeO/hg-C_3_N_4_ was almost 100% within 60 min, far beyond the hg-C_3_N_4_, BiFeO_3_, and 10BiFeO/bg-C_3_N_4_ catalysts. The apparent discrepancies in photocatalytic performance will be discussed from micro-nano structure, adsorption properties, and the formation of the internal field.

#### 3.3.1. Micro-Nano Structure

The large specific surface area of the prepared 10BiFeO/hg-C_3_N_4_ and well-dispersed BiFeO_3_ particles on the hg-C_3_N_4_ support (Figure 7) increased the contact probabilities between the reactants and the catalysts. In addition, the unique honeycomb morphology of 10BiFeO/hg-C_3_N_4_ formed multiple reflections of light through interlaminar pore structure. This multi-reflection can effectively improve the utilization of light energy and promote the diffusion of the reactants [36].

#### 3.3.2. Adsorption of Reactant Molecules on the Catalysts

In the typical heterogeneous catalytic degradation of RhB, the key step is the adsorption of the reactant molecules onto the catalyst’s surface. The RhB solution with photocatalysts was stirred in a dark room for 30 min to achieve a stable absorption–desorption equilibrium before light irradiation. The UV/vis absorption of the dye solution was used to record the amount of the reactant molecules that were adsorbed on the surface of the catalysts in the dark. As shown in Figure 8a, the capacity of the adsorption molecules for hg-C_3_N_4_ and bg-C_3_N_4_ was similar. The adsorption ability was enhanced by a factor of approximately seven when BiFeO_3_ particles were deposited on graphite C_3_N_4_. The enhancement indicated that the strong Stern layers with the dye molecules formed on the surface of BiFeO_3_ ferroelectrics [44]. As shown in Figure 8b, the spontaneous polarization of ferroelectrics can induce surface macroscopic charges. The induced charges could be compensated by the charged species from the external environment [44]. It has been reported that the polarized ferroelectrics showed stronger adsorption of dye molecules on C^+^ and C^−^ surfaces than the non-ferroelectrics surface. This interaction between the dipole moment of a polar molecule (or induced dipole moment) and the polarization of ferroelectric domains can promote the bond-breaking of the adsorbed molecules and enhance photocatalytic activity [45,46,47].

The degradation of anionic dye methyl red (MR) was carried out following the same reaction conditions as the degradation of RhB to further identify the importance of the formation of stern layers with chemisorbed molecules. The adsorption of methyl red onto the catalysts was not evident (Appendix A). Appendix A showed a 19% difference in MR degradation rate between the hg-C_3_N_4_ and 10BiFeO/hg-C_3_N_4_ within 60 min, and this difference is lower than that in RhB degradation. This indicated that the adsorption of RhB onto the catalysts played a vital role. Despite the lack of dye adsorption, 10BiFeO/hg-C_3_N_4_ still shows the highest activity in the photodegradation of methyl red among all tested catalysts.

#### 3.3.3. The Internal Field

The separation efficiency of the photogenerated electron–hole pairs significantly impacted the photocatalytic activity. The fluorescence spectra and photoelectrochemical experiments were carried out to study the charge transfer resistance and the separation efficiency of the charge carriers. The room temperature PL spectra with an excitation wavelength of 325 nm for the hg-C_3_N_4_, 10BiFeO/hg-C_3_N_4_, and 10BiFeO/bg-C_3_N_4_ were displayed in Figure 9a. The composites have much lower PL intensity than the pure samples, meaning that the recombination of photogenerated electron–hole pairs can be effectively suppressed. The PL intensity of 10BiFeO/hg-C_3_N_4_ was weaker than that of 10BiFeO/bg-C_3_N_4_, which can be attributed to the special nanoarchitecture. The larger surface area and thinner thickness of honeycomb-like g-C_3_N_4_ promoted charge separation and migration [36]. Meanwhile, the efficient formation of heterojunction originated from the strong interaction between the well-dispersed BiFeO_3_ particles and honeycomb-like g-C_3_N_4_. The strong interaction contributes to the inhibition of photogenerated electron–hole recombination. The charge separation dynamics of the hg-C_3_N_4_, 10BiFeO/hg-C_3_N_4_, and 10BiFeO/bg-C_3_N_4_ were investigated by the time-resolved photoluminescence spectra (Figure 9b). In addition, the photocurrent–time response curves and EIS curves for the samples indicated that the 10BiFeO/hg-C_3_N_4_ had improved charger carrier mobility due to its highest current density (Figure 9c) and smallest charge transfer resistance according to the smallest hemicycle radius (Figure 9d) [39,48].

We employed UV-vis absorption spectra to examine the optical absorption properties of the synthesized pure graphite C_3_N_4_, BiFeO_3_, and the composite samples. As shown in Figure 10a,b, the absorption peaks of the pure graphite C_3_N_4_ and BiFeO_3_ were well consistent with the previous literature [22], and the absorption edge was located at around 450 nm and 600 nm, respectively. The absorption peak intensity for the hg-C_3_N_4_ was stronger than bg-C_3_N_4_ in the ultraviolet and visible light region due to the large surface area and the multiple reflections of incident light in the honeycomb structure [36]. The 10BiFeO/hg-C_3_N_4_ presented stronger peak intensity than 10BiFeO/bg-C_3_N_4_ attributed to the same reason, as shown in Figure 10b.

The band gap was calculated according to the following Equation (1) [46]:(αhν)^n^ = k(hν − E_g_) (1)
where h is Planck’s constant, ν is vibration frequency, α represents the absorption coefficient, E_g_ is the band gap, and k is the proportional constant. Figure 10c suggests that the band gap of the BiFeO_3_ and hg-C_3_N_4_ were 2.13 eV and 2.78 eV, respectively. Based on the band gap, we further accessed the conduction band (CB) and valence band (VB) levels of the BiFeO_3_ and hg-C_3_N_4_ by using the Mulliken electronegativity formula [49]. The CB and VB for the BiFeO_3_ were estimated to be +2.63 eV and +0.50 eV (vs. NHE), and the CB and VB of the hg-C_3_N_4_ were +1.63 eV and −1.15 eV, respectively (Figure 10d). The absorption edge of the composites was well aligned with the pure graphite C_3_N_4_ and BiFeO_3_. This indicated that the band gap of the graphite C_3_N_4_ and BiFeO_3_ in the composites was almost unchanged [28].

DMPO spin-trapping ESR was carried out to identify the possible reactive species involved in the degradation of RhB. The free radicals generated by BiFeO_3_, hg-C_3_N_4_, and 10BiFeO/hg-C_3_N_4_ were captured under visible light illumination. In Figure 11a, the ESR spectra of BiFeO_3_ indicated that only •OH radicals could be formed during the photoreaction. Considering the band structure of BiFeO_3_, the holes from BiFeO_3_ have the sufficient capacity to oxidize OH/H_2_O into •OH radicals due to its more positive VB potential (2.63 eV vs. NHE) than E (OH/•OH) (1.99 eV vs. NHE) and E(H_2_O/•OH) (2.38 eV vs. NHE) [50]. The signal of DMPO-•OH can also be observed in the hg-C_3_N_4_, and the •OH may come from the H_2_O_2_ produced by the reaction of O_2_ and H_2_O [51]. Meanwhile, the more negative CB in the hg-C_3_N_4_ can directly reduce O_2_ to form O_2_^−^. Regarding the 10BiFeO/hg-C_3_N_4_, both the signals of DMPO-•OH and DMPO-O_2_^−^ can be clearly observed. Furthermore, to rule out the possibility that DMPO-•OH originated from the degradation of the adduct DMPO-•OOH between DMPO and O_2_^−^ [52,53], we investigate the degradation of coumarin to detect hydroxyl radicals preliminarily. As shown in Appendix A, for the 10BiFeO/hg-C_3_N_4_, the luminescence intensity of coumarin decreased rapidly with irradiation time due to the formation of •OH radicals, which can well support the results of ESR spin trapping experiments. Based on the possible reactive species, tertbutanol (t-BuOH), benzoquinone (BQ), and triethanolamine (TEOA) were chosen as quenchers of •OH radical, O_2_^−^ radical, and h^+^, respectively, to investigate the role of active radicals on the degradation rate of RhB over 10BiFeO/hg-C_3_N_4_ [20]. From Figure 11b, the photodegradation rate has no noticeable change after introducing t-BuOH and TEOA into the reaction system. While after the addition of BQ, the photodegradation was significantly inhibited, suggesting that the O_2_^−^ radical was the critical active species.

The proper band alignment between hg-C_3_N_4_ and BiFeO_3_ led to the formation of an efficient heterostructure and facilitated charge separation. Two mainstream charge-transfer mechanisms, type-II and Z-scheme, have been widely acknowledged [19]. For the type-Ⅱ mode, the two coupled semiconductors generate electron and hole pairs under irradiation. The photogenerated electrons were transferred from hg-C_3_N_4_ to BiFeO_3_, while the photogenerated holes were moved in the opposite direction. The trapping experiments suggested that O_2_^−^ radicals are the main reactive species in the degradation reaction. The accumulated electrons in the BiFeO_3_ CB cannot generate O_2_^−^ radicals because of its more positive CB edge potential than the reduction potential of O_2_. Therefore, it is hard to explain the mechanism of this reaction using the predicted type-II heterojunction. Regarding the Z-scheme mode, the photogenerated electrons in the BiFeO_3_ CB directly recombined with the photogenerated holes. The reserved photogenerated electrons in the CB of hg-C_3_N_4_ and holes in the VB of BiFeO_3_ still maintain the strong redox ability. The accumulated electrons in hg-C_3_N_4_ CB have enough energy to generate O_2_^−^ radicals, effectively overcoming the limitation in type-Ⅱ mode.

The photo-deposition of PbO_2_ and Ag particles on the 10BiFeO/hg-C_3_N_4_ was carried out to gain further insight into the transfer route of the photogenerated carriers. In general, PbO_2_ nanoparticles characterize the site of the photogenerated hole flow, and Ag can reveal the photogenerated electron route [54,55]. As shown in Figure 12, according to the lattice fringe spacings from the plane of different species, the PbO_2_ nanoparticles were primarily photo-deposited onto the BiFeO_3_ particles, and the Ag nanoparticles are mainly distributed on hg-C_3_N_4_. The different distribution routes demonstrated that the photogenerated holes reserved in BiFeO_3_ and the electrons tend to remain in hg-C_3_N_4_. In situ irradiated XPS spectra were also performed to reveal the charge transfer between hg-C_3_N_4_ and BiFeO_3_. Upon exposure to photon illumination, C 1s and N 1s peaks move to lower binding energy, while the Bi 4f and Fe 2p peaks move in the opposite direction. The moving directions suggested a charge transfer from BiFeO_3_ to hg-C_3_N_4_ under light irradiation (Figure 13) [56]. These results provided convincing evidence that the Z-scheme mode is the proper reaction mechanism for RhB photocatalyzed degradation over 10BiFeO/hg-C_3_N_4_.

Overall, a possible mechanism for the photodegradation of RhB over 10BiFeO/hg-C_3_N_4_ can be deduced and shown in Figure 14. Under visible irradiation, the excited electrons on the BiFeO_3_ CB migrated to the hg-C_3_N_4_ VB on the interface between BiFeO_3_ and hg-C_3_N_4_. As a result, it left a trail of electrons and holes on the hg-C_3_N_4_ CB and BiFeO_3_ VB. The interfacial field was built within the interface of two regions, substantially increasing the separation efficiency of the charge carrier. This Z-scheme heterojunction exhibited both high charge separation and remarkable redox ability. The internal electric field, caused by the spontaneous polarization of BiFeO_3_ species, can increase the band bending and facilitate the migration of photogenerated carriers in the opposite direction. The superposition of these two fields can make the photogenerated carriers separate in space and effectively reduce the recombination opportunity. In addition, the ferroelectricity of BiFeO_3_ ensures a tightly bound layer of the dye molecule. The tightly bound layer can reduce the required energy to break bonds and enhance photocatalytic performance.

## 4. Conclusions

In this study, we synthesized a novel 10BiFeO/hg-C_3_N_4_ type-Z heterojunction photocatalyst via a simple, economical, non-toxic, and energy-saving one-step method. The characterization results showed that the BiFeO_3_ particles are well dispersed on the honeycomb-like g-C_3_N_4_, and the close contact led to the strong interaction between the two species. All of the composites showed a stronger photocatalytic degradation capacity to RhB than pure graphite C_3_N_4_ and BiFeO_3_ under the same conditions. The 10BiFeO/hg-C_3_N_4_ presented the best catalytic performance. The mechanism with respect to the photocatalytic performance improvement over the composites was discussed in detail. The unique honeycomb-like morphology of 10BiFeO/hg-C_3_N_4_ promoted the reactant diffusion and utilization of light energy. The spontaneous polarization of ferroelectric BiFeO_3_ induced a greater degree of binding between the polar RhB cation and the catalyst surface. Moreover, the synergistic effect between ferroelectric polarization and direct Z-scheme heterojunction leads to the efficient separation and the transfer of photogenerated carriers.

## Figures and Tables

**Figure 1 nanomaterials-12-03970-f001:**
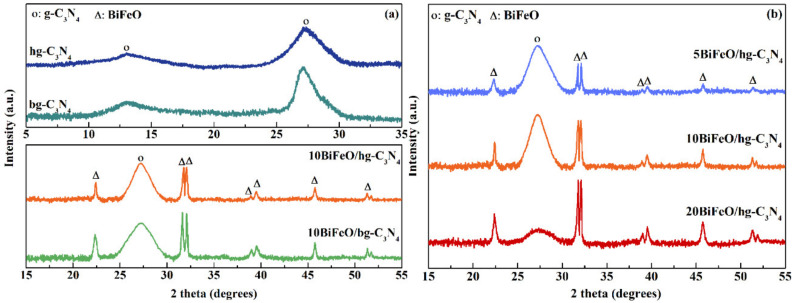
XRD patterns of following catalysts: (**a**) hg-C_3_N_4_, bg-C_3_N_4_, 10BiFeO/hg-C_3_N_4_, and 10BiFeO/bg-C_3_N_4_; (**b**) *x*BiFeO/hg-C_3_N_4_ (5%, 10%, and 20% BiFeO_3_).

**Figure 2 nanomaterials-12-03970-f002:**
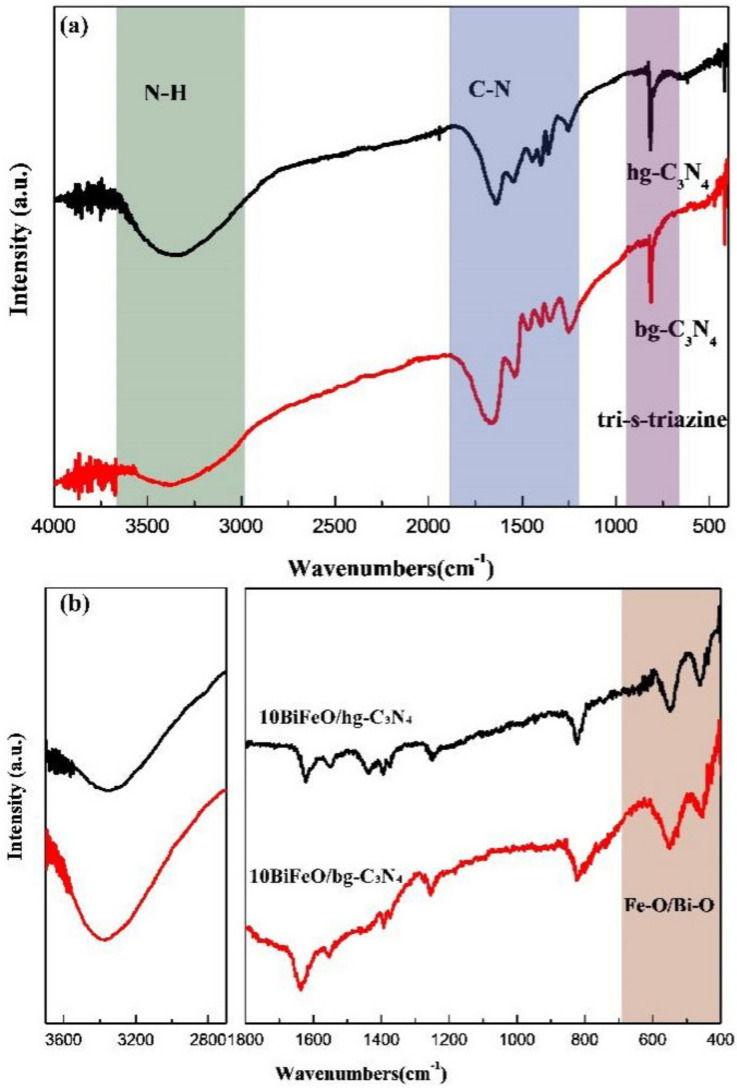
FT-IR transmission spectra of (**a**) hg-C_3_N_4_ and bg-C_3_N_4_; (**b**) 10BiFeO/hg-C_3_N_4_, and 10BiFeO/bg-C_3_N_4_.

**Figure 3 nanomaterials-12-03970-f003:**
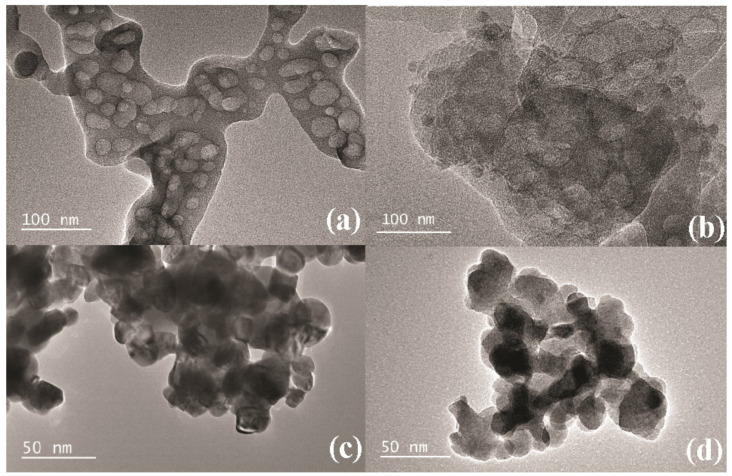
TEM images of synthesized samples: (**a**) bg-C_3_N_4_; (**b**) hg-C_3_N_4_; (**c**) 10BiFeO/bg-C_3_N_4_; and (**d**) 10BiFeO/hg-C_3_N_4_.

**Figure 4 nanomaterials-12-03970-f004:**
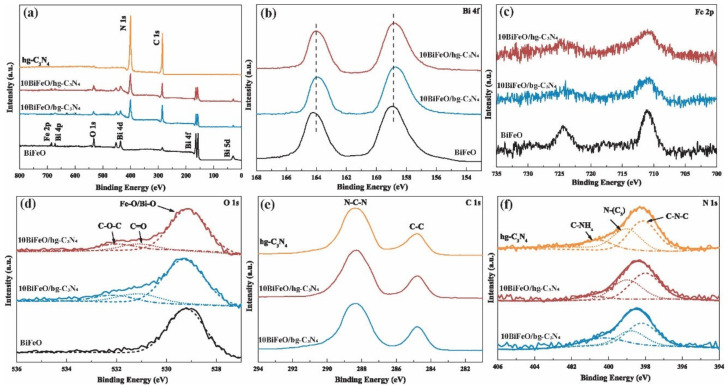
XPS spectra profiles (**a**) survey of hg-C_3_N_4_, 10BiFeO/hg-C_3_N_4_, 10BiFeO/bg-C_3_N_4_, and BiFeO; (**b**) Bi 4f of 10BiFeO/hg-C_3_N_4_, 10BiFeO/bg-C_3_N_4_, and BiFeO; (**c**) Fe 2p of 10BiFeO/hg-C_3_N_4_, 10BiFeO/bg-C_3_N_4_, and BiFeO; (**d**) O 1s of 10BiFeO/hg-C_3_N_4_, 10BiFeO/bg-C_3_N_4_, and BiFeO; (**e**) C 1s of hg-C_3_N_4_, 10BiFeO/hg-C_3_N_4_, and 10BiFeO/bg-C_3_N_4_; (**f**) N 1s of hg-C_3_N_4_, 10BiFeO/hg-C_3_N_4_, and 10BiFeO/bg-C_3_N_4_.

**Figure 5 nanomaterials-12-03970-f005:**
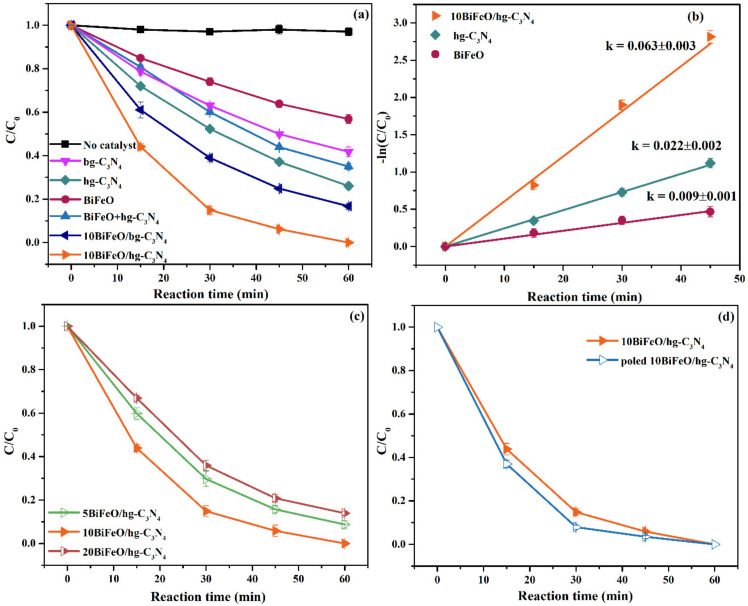
Photocatalytic degradation of RhB: (**a**) the synthesized g-C_3_N_4_, BiFeO and BiFeO/g-C_3_N_4_ samples; (**b**) the pseudo-first-order kinetics curves of BiFeO, hg-C_3_N_4_ and 10BiFeO/hg-C_3_N_4_; (**c**) *x*BiFeO/hg-C_3_N_4_ (5%, 10%, and 20% BiFeO_3_); (**d**) electrically poled 10BiFeO/hg-C_3_N_4_.

**Figure 6 nanomaterials-12-03970-f006:**
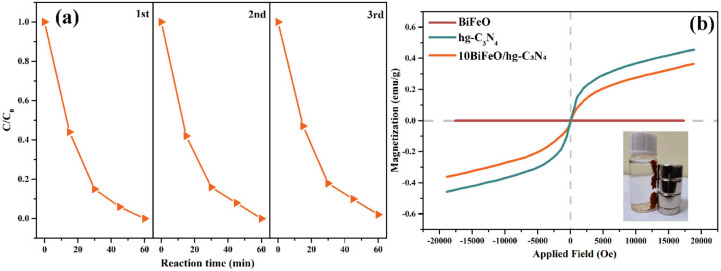
(**a**) Recycling runs for the photocatalytic RhB in the presence of 10BiFeO/hg-C_3_N_4_ under visible light irradiation; (**b**) M-H hysteresis loop of 10BiFeO/hg-C_3_N_4_ after the photocatalytic measurements.

**Figure 7 nanomaterials-12-03970-f007:**
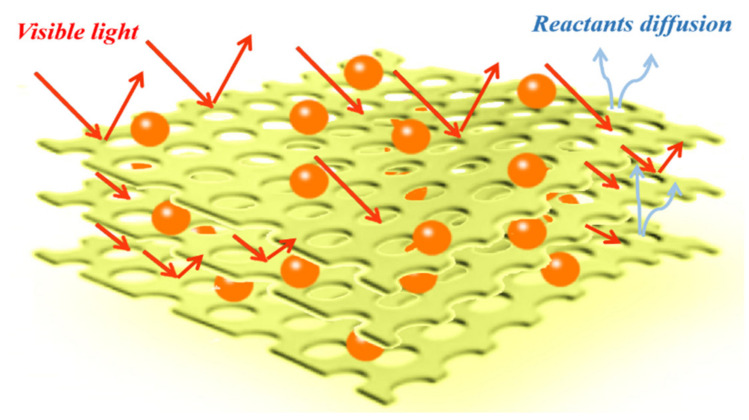
Schematics of honeycomb-like morphology.

**Figure 8 nanomaterials-12-03970-f008:**
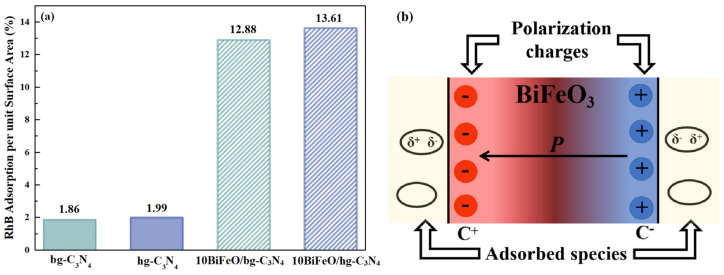
(**a**) Adsorption of RhB by g-C_3_N_4_ and BiFeO/g-C_3_N_4_ under dark conditions for 30 min; (**b**) internal polarization and adsorption mechanisms.

**Figure 9 nanomaterials-12-03970-f009:**
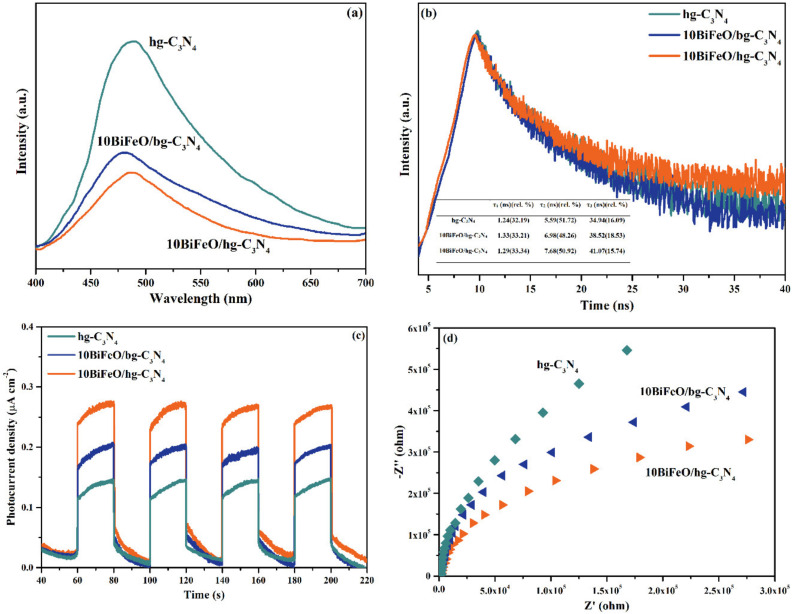
Steady-state PL spectra (**a**), time-resolved PL spectra (**b**), photocurrent response density (**c**) and EIS Nyquist plots (**d**) of hg-C_3_N_4_,10BiFeO/hg-C_3_N_4_ and 10BiFeO/bg-C_3_N_4_.

**Figure 10 nanomaterials-12-03970-f010:**
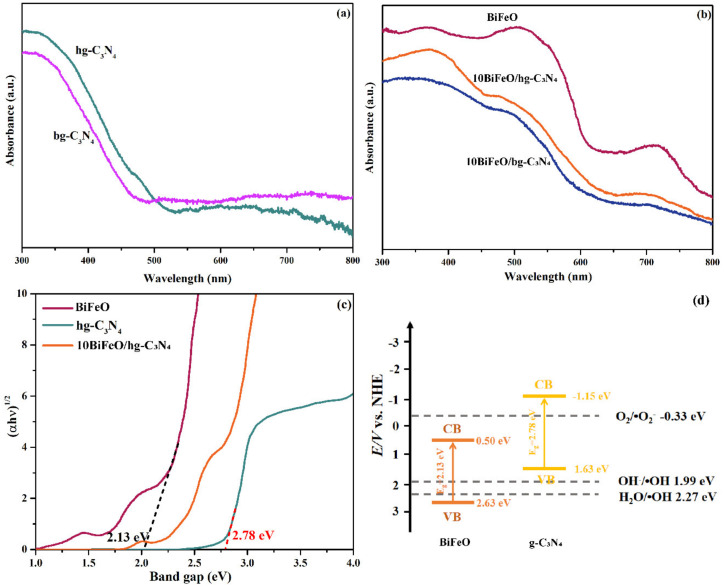
(**a**) UV-visible spectra of hg-C_3_N_4_ and bg-C_3_N_4_; (**b**) UV-visible spectra of BiFeO, 10BiFeO/hg-C_3_N_4_ and 10BiFeO/bg-C_3_N_4_; (**c**) estimated band gaps of BiFeO, hg-C_3_N_4_ and 10BiFeO/hg-C_3_N_4_; (**d**) band gap structure of 10BiFeO/hg-C_3_N_4_.

**Figure 11 nanomaterials-12-03970-f011:**
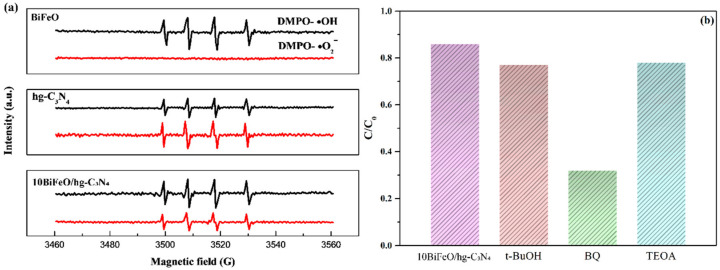
(**a**) ESR signals of BiFeO, hg-C_3_N_4_ and 10BiFeO/hg-C_3_N_4_; (**b**) Photocatalytic activity of 10BiFeO/hg-C_3_N_4_ for the degradation of RhB in the presence of different scavengers.

**Figure 12 nanomaterials-12-03970-f012:**
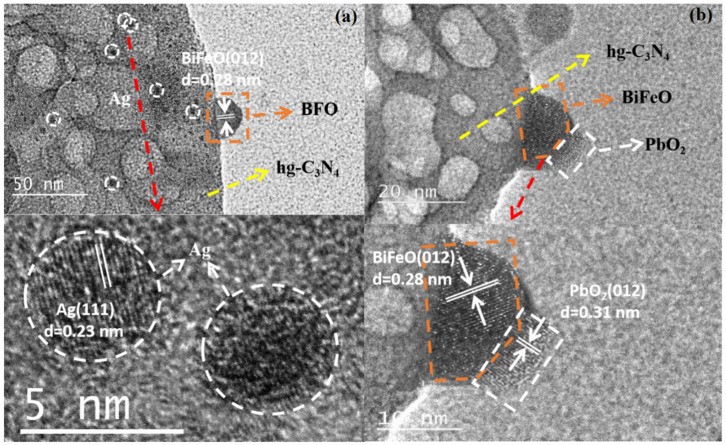
HRTEM images of (**a**) Ag and (**b**) PbO_2_ photo deposited in the existence of a BiFeO/hg-C_3_N_4_ sample, respectively.

**Figure 13 nanomaterials-12-03970-f013:**
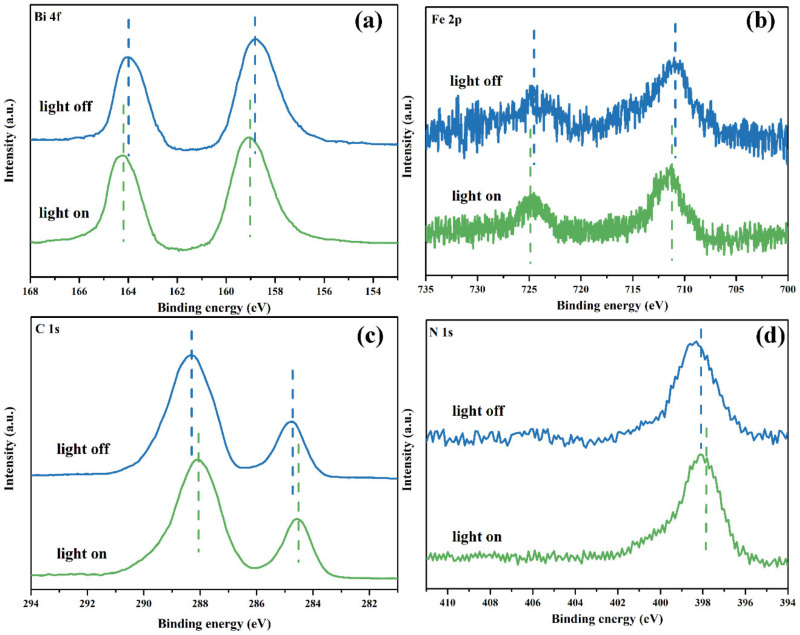
XPS spectra of BiFeO/g-C_3_N_4_ in the dark and light irradiation: (**a**) Bi 4f, (**b**) Fe 2p, (**c**) C 1s, (**d**) N 1s.

**Figure 14 nanomaterials-12-03970-f014:**
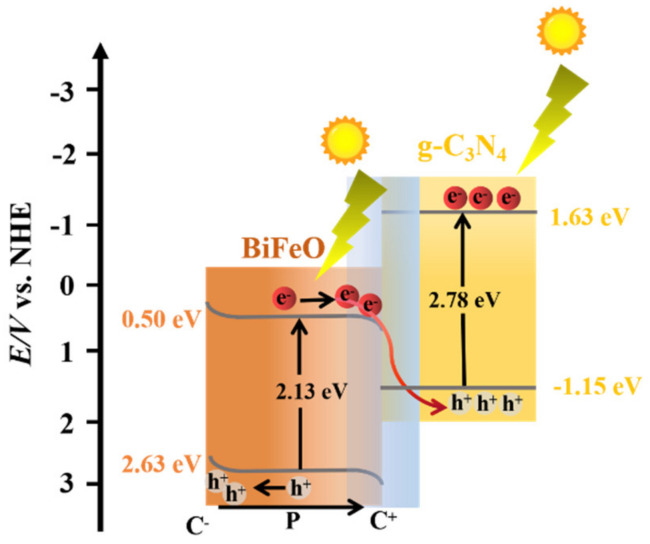
Schematic illustration of the charge-carrier migration mechanism of Z-scheme heterojunction for 10BiFeO/g-C_3_N_4_.

**Table 1 nanomaterials-12-03970-t001:** Structural and textural properties of the catalysts.

Sample	BiFeO Content (wt. %) ^a^	D_BiFeO_ (nm) ^b^	A_BET_ (m^2^/g) ^c^	V_Total_ (cm^3^/g) ^c^
bg-C_3_N_4_	-	-	38.8	0.11
hg-C_3_N_4_	-	-	61.4	0.19
10BiFeO/bg-C_3_N_4_	10.7	29.3	18.2	0.07
10BiFeO/hg-C_3_N_4_	9.2	18.5	41.8	0.12
5BiFeO/hg-C_3_N_4_	4.3	12.4	49.6	0.14
20BiFeO/hg-C_3_N_4_	21.5	36.8	22.7	0.09

^a^ The content of BiFeO_3_ in the catalysts were detected by ICP-AES; ^b^ The mean crystallite size of BiFeO_3_ calculated from XRD; ^c^ The BET surface area, pore size and volume were determined by N_2_ physical adsorption.

## Data Availability

The data presented in this study are available in this article.

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
