# Peer review of "Facilitated Photocatalytic Degradation of Rhodamine B over One-Step Synthesized Honeycomb-Like BiFeO3/g-C3N4 Catalyst"

_nanomaterials, 2022, doi:10.3390/nano12223970_

Round 1

Reviewer 1 Report

In this work, Authors explored the properties of a composite BiFeO3/g-C3N4 catalyst and its activity in the photocatalytic degradation of Rhodamine B.

Very recently, Mohanty et al. published on Inorganic Chemistry Communications the paper entitled: “Enhanced photocatalytic degradation of rhodamine B and malachite green employing BiFeO3/g-C3N4 nanocomposites: An efficient visible-light photocatalyst” 2022, 138, 109286.

Even though the synthesis of the composite materials is different, the contents of the two papers are too similar and, as it is, the work under question loses its originality. So, the paper is not publishable in this version. However, since a lot of work is described and some results are of interest, I suggest to the Authors to modify and deeply revise the paper, focussing on the comparison between 10BFO/hg-C3N4, 5BFO/hg-C3N4, 20BFO/hg-C3N4. The material 10BFO/hb-C3N4 can be useful for comparing “hg” with “hb”. Parent components should be completely eliminated from the discussion. In consequence, FT-IR (figure 2), TEM (Figure 3), XPS (Figure 4) etc…and the relative discussion should be rewritten.

If the decision of the Authors is to follow my suggestion, I list in the following some points that Authors need to address in the new revised version:

1)      Check all the cited references: for example, in the Introduction refs 6, 7 do not concern “biochemical processes”

2)      In the Experimental a lot of details are missing. For example, Authors talk about photoelectrochemical and impedance measurements but no explanation about how they did the experiment is present. Every technique used for the discussion must be presented in detail in the Experimental Section.

3)      To avoid confusion, I suggest abbreviating BiFeO3 as BiFeO (and not BFO): in fact, B and F are other elements of the periodic table. Moreover, Authors got wrong by writing BFO in the text, BiFO in the pictures…..so enhancing the confusion.

4)      Table 1 reports the textural properties of 10BFO/hg-C3N4, 5BFO/hg-C3N4, 20BFO/hg-C3N4. However, at page 6 any comment about SSA and porosity as a function of wt% of BFO is present

5)      XPS discussion: 5BFO/hg-C3N4, 20BFO/hg-C3N4 are missing. As suggested, Authors should delete all data of parent compounds (unless they are strictly necessary for the discussion). More supporting literature references should be introduced.

6)      Par 3.2 page 7: Authors write: “the better performance of hg-C3N4 mainly resulted from the higher photoenergy efficiency, reduced surface defects, and enhanced charge separation”. How did they demonstrate it?

Moreover: “the formation of the heterojunction…..” How did they demonstrate it?

7)      Figure 5: error bars must be included. I suppose that each experimental point has been confirmed at least once. Error bars allow to eventually claim whether the differences between the two trends of Figure 5d are statistically different or not.

8)      Par 3.3.1 Authors write: “the large specific surface area of prepared 10BFO/hg-C3N4 …….increases the contact probabilities between reactants and the catalysts”. Data from Table 1 are in contrast with this sentence: in fact, 5BFO/hg-C3N4 has a SSA of 49.6 m2/g and 10BFO/hg-C3N4  has a smaller SSA 41.8 m2/g and it is less performant (Figure 5c). Could you explain the result in a coherent way?

9)      In Figure 8 the adsorption of RhB is shown. I don’t understand how Authors take into account the adsorption in Figure 5a. Which is the contribute of photocatalytic degradation and of adsorption of RhB at the different irradiation times of Figure 5a?

10)  Figure 9a: PL spectra. As the optical properties are strongly dependent on particle size, the differences reported in the Figure are comparable only if particle sizes of the emitting materials are comparable. No data about granulometry is present in the paper and should be reported.

11)  Figure 9b: t2 lifetimes are very close to each other. What is the confidence interval of this set of measures? This information is mandatory to establish whether these small differences are significant or not.

12)   Figure 11: ESR spin trapping experiments. Literature data report that [DMPO–OH]• species may originate from degradation of the adduct [DMPO–OOH]• between DMPO and O2 (E. Finkelstein, G.M. Rose, E.J. Rauckman, Mol. Pharmacol. 21 (1982) 262–265;  G.R. Buettner, Free Radic. Res. Commun. 19 (1993) S79–S87). I suggest revealing the formation of OH radicals by using terephthalic acid or coumarin, whose hydroxylated products are fluorescent (H. Czili, A. Horvath, Appl. Catal. B: Environ. 81 (2008) 295–302).

13)  Page 15. In the discussion of the proposed mechanism, the direction of transfer of excited electrons is in the opposite sense with respect to that shown in Figure 14.

Reviewer 2 Report

The authors in this paper have successfully synthesized a novel BiFeO3/hg-C3N4 type-Z heterojunction photocatalyst via a simple energy-saving one-step method. The characterization results showed that the BiFeO3 particles are well dispersed on a honeycomb-like g-C3N4. All the composites showed a stronger photocatalytic degradation of Rhodamine B dye than pure graphite C3N4 and BiFeO3 under the same conditions. The honeycombed BiFeO3/hg-C3N4 presented the best catalytic performance.

The field of photocatalysis over the last 20 years has expanded enormously where chemical compositions and morphological properties are often a nightmare to discern novelty and activity enhancements. within its own context the paper as presented here is a useful extension of composition and morphological properties in this regard. Its nicely presented and the data underpins the outcomes well and the discussions and interpretations make sound interpretations with the current literature. The study here makes a very deep analytical evaluation and description of the honeycombed catalyst. It's always difficult to assess catalytical activity since there is no standard for comparison other than the separate breakdown components of the synthesised catalyst. Comparative evaluations with a more re-known catalyst like for example a nano-titania would have been useful.

However, there are some research articles in the literature very closely related to the outcomes in this study some listed below and not mentioned in the paper. I feel the authors should, therefore, take into consideration similar works and discern the advancements made in their work.

Bismuth Iron Oxide Nanocomposite Supported on Graphene Oxides as the High Efficient, Stable and Reusable Catalysts for the Reduction of Nitroarenes under Continuous Flow Conditions

  • November 2016
  • Chemical Engineering Journal 314
DOI:10.1016/j.cej.2016.11.128  

Journal of Materials Research and Technology

Volume 8, Issue 6, November–December 2019, Pages 6375-6389  

Critical review: Bismuth ferrite as an emerging visible light active nanostructured photocatalyst

Author links open overlay panelSyedIrfanabZhengZhuanghaoabFuLiaYue-XingChenaGuang-XingLiangaJing-TingLuoaFanPingab

Round 2

Reviewer 1 Report

see the uploaded file
